# Interspecific and Environmental Influence on the Foliar Metabolomes of *Mitragyna* Species Through Recursive OPLSDA Modeling

**DOI:** 10.3390/plants14172721

**Published:** 2025-09-01

**Authors:** Tushar Andriyas, Nisa Leksungnoen, Suwimon Uthairatsamee, Chatchai Ngernsaengsaruay, Sanyogita Andriyas

**Affiliations:** 1Department of Forest Biology, Faculty of Forestry, Kasetsart University, Bangkok 10900, Thailand; thugnomics28@gmail.com (T.A.); fforsmu@ku.ac.th (S.U.); 2Center for Advance Studies in Tropical Natural Resources, National Research University, Bangkok 10900, Thailand; 3Kasetsart University Research and Development Institute (KURDI), Kasetsart University, Bangkok 10900, Thailand; 4Department of Botany, Faculty of Science, Kasetsart University, Bangkok 10900, Thailand; fsciccn@ku.ac.th; 5Department of Irrigation and Drainage Engineering, Vaugh Institute of Agriculture Engineering and Technology, Sam Higginbottom University of Agriculture, Technology, and Sciences, Prayagraj 211007, India; sandriyas@gmail.com

**Keywords:** *Mitragyna* species, multiclass modeling, clustering, untargeted metabolomics

## Abstract

Understanding interspecific and environmental influences on secondary metabolite profiles can be critical in plant metabolomics. This study used a hierarchical orthogonal projections to latent structure discriminant analysis (OPLS-DA) to classify the foliar metabolomes of four naturally growing *Mitragyna* species in Thailand, *M. speciosa*, *M. diversifolia*, *M. hirsuta*, and *M. rotundifolia*. Using a recursive binary classification, interspecific and environmental influences were determined in multiple class separations, while identifying key metabolites driving these distinctions. Gas chromatography–mass spectrometry (GC-MS) annotated 409 metabolites, and through a progressive class differentiation using hierarchical OPLS-DA, *M. speciosa* exhibited a metabolome distinct from the other three species. However, the metabolomes of *M. hirsuta* and *M. rotundifolia* had a lot of overlap, while *M. diversifolia* displayed regional metabolic variation, emphasizing the role of environmental factors in shaping its chemical composition. Key metabolites, such as mitragynine, isorhynchophylline, squalene, and vanillic acid, among others, were identified as major discriminators across the hierarchical splits. Unlike conventional OPLS-DA, which struggles with multiclass datasets, the recursive approach identified class structures that were biologically relevant, without the need for manual pairwise modeling. The results aligned with prior morphological and genetic studies, validating the method’s robustness in capturing interspecific and environmental differences, which can be used in high-dimensional multiclass plant metabolomics.

## 1. Introduction

Various species from the *Mitragyna* are found across Africa and Asia, with *Mitragyna diversifolia*, *M. hirsuta*, *M. rotundifolia*, and *M. speciosa* (kratom) occurring frequently in Thailand [1]. These species are known to contain a diverse range of secondary metabolites, including alkaloids, polyphenols, flavonoids, triterpenoids, triterpenoid saponins, monoterpenes, and secoiridoids [2]. Metabolites, which include mitragynine and mitraversine, find extensive use in ethnic treatments of cough, fever, diarrhea, and malaria [2,3]. Additionally, potent antioxidant, antimicrobial, anticancer, and enzyme-inhibitory properties have also been reported for these metabolites, highlighting their significance in pharmaceutical and therapeutic research [4,5].

Both genetic and environmental factors can influence the quality and quantity of secondary metabolites in plants. Razafimandimbison and Bremer [6] indicated that *M. rotundifolia* shares genetic similarities with *M. diversifolia*, an Asian species, as well as *M. inermis*, which is only found in the Sudanese region of Africa. Additionally, *M. rotundifolia* was also reported to be genetically similar to *M. speciosa* and has been used as a substitute for *M. speciosa* in traditional ethnopharmaceutical applications [7]. Beyond genetic influence, abiotic conditions, particularly light intensity and water availability, have been shown to significantly impact the total alkaloid accumulation in kratom. A recent study demonstrated that variations in these factors can induce metabolic shifts, further highlighting the interplay between genetic heritage and environmental adaptability in shaping the secondary metabolome [8]. Consequently, the secondary metabolite profiles may exhibit variations that are influenced at both interspecific and regional levels, invoking the need for precise methodologies to characterize such differences in metabolic composition.

Plant metabolomics, combined with analytical techniques such as untargeted profiling, is used to indicate the presence of secondary metabolites, which can be used further to understand the accumulation and underlying pathways of their production. Common methods include gas chromatography–mass spectrometry (GC-MS), liquid chromatography–mass spectrometry (LC-MS), and nuclear magnetic resonance (NMR) [9]. GC-MS offers high-resolution separation, reproducible retention times, and cost efficiency, making it advantageous over LC-MS for analyzing volatile and thermally stable compounds [10]. However, its use in derivatization for non-volatile metabolites adds complexity and variability [11,12]. Additionally, GC-MS systems require frequent maintenance and calibration, limiting their application in large-scale metabolomic studies, despite their value in analyzing complex biological systems.

Such spectrometry approaches result in complex datasets and need a comprehensive workflow that integrates unsupervised and supervised analytical approaches to extract biologically meaningful clusters [13]. Unsupervised methods, such as principal component analyses (PCA) or hierarchal clustering, and supervised techniques, such as partial least squares discriminant analyses (PLS-DA) and orthogonal PLS-DA (OPLS-DA), are used for dimensionality reduction, biomarker identification, and predictive modeling [13,14]. With the increasing complexity of these datasets, which can have multiple classes, multi-class approaches can be used to optimize separation and maintain statistical rigor and interpretability. Constructing one-vs.-one OPLS-DA models for multiple classes remains effective, but can be time-consuming and labor-intensive, especially when dealing with several classes.

Automated multiclass approaches can lead to improved efficiency but still face limitations in their flexibility and interpretability [15,16]. Integrating hierarchical clustering with supervised OPLS-DA provides a progressive classification to identify hierarchical relationships while preserving discriminatory power. In the present context, the untargeted foliar metabolomes of four *Mitragyna* species (*M. diversifolia*, *M. hirsuta*, *M. rotundifolia*, and *M. speciosa*) found naturally in Thailand [17,18], under diverse climatic and soil conditions, were analyzed.

The objective was to determine both the interspecific and regional differences based on the foliar secondary metabolite compositions of these species across various regions of Thailand. To address the complexity of multiclass classification of the foliar metabolome, we integrated hierarchical clustering with supervised orthogonal partial least squares or projection and a latent structures discriminant analysis (OPLS-DA) to obtain a recursive classification model. The methodological novelty lies in the use of this tool in classifying the metabolomes of the *Mitragyna* species, with the hierarchical structure mirroring phylogenetic or ecological relationships, wherein species within shared clusters could exhibit common ancestry or environmental adaptations. This approach mitigates the overfitting risks inherent in monolithic multiclass models, as well as multiple binary class models. Through this framework, we identified foliar metabolites deemed significant as interspecific and environmental discriminators across the four *Mitragyna* species, a contribution that to our knowledge addresses a previously unexamined research gap.

## 2. Results

### 2.1. Statistical Analysis of Secondary Metabolites

Gas chromatography–mass spectrometry of ethanol extracts resulted in a foliar metabolomes of 409 unique metabolites (Appendix A). The unsupervised PCA scores plot (Figure 1) indicates that *M. speciosa* formed a distinct cluster, while the clusters of samples from *M. diversifolia*, *M. rotundifolia*, and *M. hirsuta* were not well separated (shaded ellipses represent 95% confidence intervals). The model variance explained by the first two principle axes was above 40%, with 45 out of the 409 metabolites found to significantly influence the loadings along PC1 and PC2 axes. Among the 409 metabolites, a total of 10 metabolites were identified as significant discriminants, which included ajmalicine, butyl 9,12,15-octadecatrienoate, mitragynine, phloroglucinol, diformylcresol, isopaynantheine, stigmasterol, speciogynine, speciociliatin, and paynantheine. Paynantheine and mitragynine positively loaded PC2 (Figure 1) and were present abundantly in *M. speciosa* (see Appendix A), while butyl 9,12,15-octadecatrienoate, which had a higher abundance in *M. diversifolia*, negatively loaded PC1. The presence of stigmasterol, which equally loaded positive PC1 and negative PC2, was elevated in *M. speciosa* relative to other species. This was in contrast to speciogynine, which had a higher presence in *M. speciosa*, and equally loaded PC1 (negative loading) and PC2 (positive loading).

The hierarchical clustering analysis was performed using Ward’s linkage method, with six broad metabolomic clusters observed (Figure 2). Similar to the clustering seen in the PCA score plot, samples from *M. speciosa* (MS) formed a coherent cluster regardless of regional origin, while the foliar metabolomes of the other three species (*M. diversifolia*, *M. hirsuta*, and *M. rotundifolia*) exhibited inter-regional divergence. The results suggest that clustering of the metabolomic structure was affected by both species and geographical presence.

### 2.2. Characterization of Mitragyna Foliar Metabolome Using Binary OPLSDA

Appendix A presents the results of the pairwise OPLS-DA, based on multiples models that analyzed samples based on species (*M. diversifolia*, *M. rotundifolia*, *M. hirsuta*, and *M. speciosa*) and geographical location (north and south) combinations. For the current multiclass dataset, this traditional analysis required exhaustive pairwise binary OPLS-DA comparisons across all group combinations, leading to considerable computational burden and increased time complexity. Additionally, only two models (*M. diversifolia* north vs. *M. diversifolia* south and *M. hirsuta* south vs. *M. rotundifolia* north) were significant, as highlighted in green in Appendix A. Such a performance indicated that there were redundant structures in the metabolomes and a need for class structure that had both better biological interpretability and was statistically significant.

### 2.3. Characterization of Mitragyna Foliar Metabolome Using Recursive OPLSDA

Based on the above proposed classes, the recursive OPLS-DA model indicated four significant splits, as can be seen in Figure 3. The classification metrics for the various splits are listed in Table 1. The first split in the OPLS-DA classification distinctly separated the metabolomes into two classes, comprising mostly of all *M. speciosa* (MS) and some *M. rotundifolia* (MR) trees in the south from the other three species, *M. diversifolia* (MD), *M. hirsuta* (MH), and *M. rotundifolia*, with three *M. rotundifolia* trees also clustered with the *M. speciosa* trees (Figure 3a).

A well-defined separation is suggestive of *M. speciosa* and *M. diversifolia* (only south) exhibiting unique foliar metabolomes compared to the rest of the dataset. Hence, the first split can be attributed to a result of interspecific differences in the foliar metabolomes. Furthermore, an intra-cluster dispersion of the *M. diversifolia*, *M. hirsuta*, and *M. rotundifolia* cluster suggests that while these species share overlapping features, internal metabolic variations may still exist. The corresponding S-plot (Figure 4a) at the first split identified metabolites responsible for this separation. A total of 141 metabolites from the 409 metabolites were deemed significant, with mitragynine, paynantheine, speciociliatine, isopaynantheine, mitraphylline, isorhynchophylline, and 5-hydroxymethylfurfural being some metabolites that characterized split 1.

The differential abundance of metabolites with previously reported pharmaceutical potential that were significant discriminators at split 1 are illustrated in Figure 5, using a univariate ANOVA with post hoc Tukey correction to determine any significant cluster-level differences (between clusters 1 and 2) in metabolite abundance (adjusted *p*-value < 0.05). As seen in Appendix A, large effect sizes (Cohen’s |d| > 0.8) also indicated substantial differences in metabolite abundance between cluster 1 and cluster 2 at split 1. The abundance of 5-hydroxymethylfurfural (d = 0.96), isorhynchophylline (1.03), and mitraphylline (0.82) was markedly reduced in cluster 1, while mitragynine (d = 2.16), speciociliatine (2.89), paynantheine (3.28), and isopaynantheine (0.70) showed elevated abundance levels.

The second split in the recursive OPLS-DA classification (Figure 3b) identified separation mainly due to locational differences. This is indicated by a clear separation between *M. hirsuta* trees in the south (MH south) and the remaining species, including *M. diversifolia* (MD north and MD south), *M. rotundifolia* (MR South), and *M. hirsuta* (MH north). Furthermore, the score plot depicts a compact clustering of MH south, suggests a distinct set of metabolic features within this subgroup, while a broader dispersion is reflective of a higher intra-class variability (interspecific metabolic heterogeneity) in MD, MR, and MH. Within the significant metabolites (112), catechol, eugenol, hexamethyl-cyclotrisiloxane, benzoic acid derivatives, vanillic acid, vitamin E, and methyl salicylate were significant in influencing the separation between the clusters identified in the second split (Figure 4b). As indicated in the Appendix A and Figure 6, the absolute effect size estimates for catechol (d = 2.10) and eugenol (0.98) were large, and along with vitamin E (0.67), these had a higher abundance in cluster 3, while the abundance of benzoic acid derivatives (d = 4.57), methyl salicylate (3.45), and hexamethyl-cyclotrisiloxane (5.13) favored cluster 4.

The resulting clusters in the third split (Figure 3c) separated most of *M. diversifolia* trees exhibiting a distinct tightly clustered group (blue ellipse) on the right, indicating a split mainly caused by interspecific differences. The presence of the *M. diversifolia* metabolome suggests that additional recursive splits could further resolve the separation. A closer look at the S-plot (Figure 4c) indicates that from 125 metabolites found significant at this split, isorhynchophylline, stigmasterol, eugenol, octadecanoic acid, mitraphylline, squalene, and phenol derivatives were significant discriminators between the two clusters obtained through hierarchal clustering. The effect sizes of these metabolites in Appendix A and Figure 7 indicate substantial differences in metabolite abundance between cluster 5 and 6. The abundance and effect size of isorhynchophylline (d = 1.06) and eugenol (1.06) were significantly higher and large in cluster 5, while octadecanoic acid (0.59) and stigmasterol (0.44) had relatively lower absolute effect sizes. Elevated abundance levels in cluster 6 were quantified for mitraphylline (d = 0.76), phenol derivatives (1.29), and squalene (0.31).

At the fourth split (Figure 3d), most of the *M. diversifolia* (MD south) samples were isolated from the remaining samples, indicating that this split was due to interspecific variations in the foliar metabolome. The remaining species, mostly constituting including *M. rotundifolia* (MR south) and *M. hirsuta* (MH north), had a broader distribution, indicating that the underlying metabolic variability could be further resolved. At the fourth split, the lowest number of metabolites (44 from 409 metabolites) was deemed significant and included ajmalicine, furan derivatives, formic acid esters, and D-allose, catechol, thymine, vitamin E, and megastigmatrienone (Figure 4d), which influenced the separation of MD south from the remaining species. As seen in Appendix A and Figure 8, large effect sizes (Cohen’s |d| > 0.8) also indicated substantial differences in metabolite abundance between clusters 7 and 8. The abundance of ajmalicine (d = 1.81), furan derivatives (effect size between 0.83–1.85), and D-allose (1.08) was significantly higher in cluster 7, while catechol (d = 1.07), megastigmatrienone (1.71), and thymine (1.38) showed elevated abundance and a large effect size in cluster 8. This progressive, refined classification reflects the stepwise identification of statistically validated metabolic differences, ensuring that each recursive split is based on meaningful biochemical variation that is either driven through interspecific variability or environmental differences due to location. Further recursive splits did not result in separation of the trees based on metabolomes, indicating that this was the final level of classification within the dataset.

## 3. Discussion

This study aimed to analyze the untargeted metabolomic profiles of four naturally growing *Mitragyna* species, *M. diversifolia*, *M. hirsuta*, *M. rotundifolia*, and *M. speciosa*, in Thailand, to determine both interspecific and regional differences in foliar secondary metabolite composition. Progressive splits through hierarchical clustering were integrated with a supervised OPLS-DA to obtain a recursive multi-class discrimination model to systematically refine class separations, while statistically validating the model at every split. Through this approach, significant metabolites responsible for splits due to interspecific and environmental differences were determined. The integration of hierarchical clustering with the supervised model allowed for an automated determination of class structures, rather than many time-consuming one-to-one binary classifiers.

A recent study using a similar approach highlighted the inherent limitations of the one-vs.-rest framework, where the OPLS-DA struggled to establish optimal decision boundaries, often resulting in suboptimal class separations and model overfitting [16]. For the Iris dataset, *Versicolor* and *Virginica* exhibited overlap, causing classification challenges in the standard OPLS-DA [16]. However, the previous hierarchical approach used an OPLS-DA through class distance computation via clustering, and then implemented binary OPLS-DA models at each split, ensuring progressive refinement in classification. The similar modified stepwise classification method in the present study resulted in a similar separation and two splits in the Iris dataset, indicating its potential in classifying multi-class metabolomic datasets. The model was then used to study the *Mitragyna* foliar metabolome.

The separation based on a combination of interspecific and geographic differences was observed in metabolomes of the four *Mitragyna* species, with the metabolomes of *M. speciosa* and *M. rotundifolia* (MR south) isolated at the first split. This indicated that the secondary metabolite composition of *M. speciosa* was markedly different from other *Mitragyna* species [19,20], particularly indicated by mitragynine and related indole alkaloids paynantheine, speciociliatine, and isopaynantheine being significant discriminators at the first split [21]. Interestingly, given that MR south was also isolated, this could be influenced by isorhynchophylline, given that its stereo isomer rhynchophylline has been reported in *M. rotundifolia* in a recent study [22]. A clear separation of *M. hirsuta* (MH south) at the second split suggests that geographical and ecological factors potentially influenced its metabolic profile. Similar observations have been reported in other plant species, where variations in seasons and elevation influenced the secondary metabolite profile [23]. The significance of benzoic acid derivatives, vanillic acid, and methyl salicylate in this separation corresponds with previous reports of these compounds influencing plant defense mechanisms and stress responses under environmental stresses [24,25].

The third split resulted in the separation of most samples of *M. diversifolia*, but the clustering of some *M. diversifolia* samples in the south in the other cluster could potentially reflect the combined influence of interspecific and environmental variations as the reason for the split. Isorhynchophylline, stigmasterol, eugenol, mitraphylline, squalene, and octadecanoic acid were identified as significant discriminators. Isorhynchophylline has been reported in the metabolome of *M. diversifolia* previously by Hama et al. [26]. A recent study identified the stereoisomer of isorhynchophylline (rhynchophylline) in *M. rontundifolia*, which was absent or in lower concentrations in other *Mitragyna* species [22]. Additionally, the presence of squalene as a significant discriminator was also observed by the same authors, who indicated its higher accumulation in *M. diversifolia* relative to the other remaining species at this split. Furthermore, the presence of sterols, such as stigmasterol and eugenol (as well as precursors such as squalene [27]), as significant discriminators may be a part of an evolutionary adaptation to stresses and maintenance of membrane-associated metabolic processes [28].

A further final split distinctly classified the metabolomes of all but one of the *M. diversifolia* trees in the south (MD south), with further recursive splitting not resulting in a biologically meaningful separation. The clustering suggested that MD south had a sufficiently unique metabolomic profile, while the remaining species share sufficient metabolic overlap to be grouped into a single cluster. This corresponds with the species being commonly found in the southern part of Thailand [26]. This led to the identification of ajmalicine, catechol, thymine, vitamin E, uncarine C, formic acid derivatives, and furan-based compounds as metabolites driving this separation. Several of these compounds have been previously identified in classifying various Rubiaceae species [29]. Several indole alkaloids, including uncarine C, have been previously identified in the metabolome of this species [26].

Previous research indicated that ajmalicine, a known monoterpenoid alkaloid [30], is found in *M. diversifolia* [31] and has been reported to accumulate under the influence of abiotic stresses [32], but it was not reported in any of the species by Sudmoon et al. [22]. Further splitting through the foliar metabolomes identified for these species suggests that the remaining metabolic variations reinforce the ability of the recursive OPLS-DA in identifying biologically meaningful separations while preventing over classification. The evaluation of the chemical profiles of these species may provide an understanding of their phylogenetic distribution in terms of secondary metabolites that are the result of adaptive and evolutionary processes that shape their response to environment [33].

The differentiation of four closely related *Mitragyna* species has been achieved using both morphological characteristics and DNA analyses, providing complementary approaches for accurate species identification. Among the four *Mitragyna* species, *M. speciosa* possesses the largest flowering heads. In contrast, *M. diversifolia* is characterized by the smallest leaves among the four species and a calyx with no lobes (subtruncate), making it readily identifiable [17]. However, morphologically, *M. hirsuta* and *M. rotundifolia* exhibit overlapping vegetative and reproductive traits, making differentiation challenging. The primary distinguishing characteristic is the calyx morphology, where *M. hirsuta* has deeply 5-lobed, spathulate–oblong or oblong calyces (1.5–2.5 mm long, with an acute apex), whereas *M. rotundifolia* has shallowly 5-lobed, triangular or semiorbicular calyces (0.3–0.5 mm long, with an obtuse or rounded apex) [17].

Sukrong et al. [7] successfully differentiated the four *Mitragyna* species in Thailand by analyzing their rDNA sequence and indicated that *M. hirsuta* is genetically closer to *M. diversifolia*, whereas *M. rotundifolia* shares greater genetic similarity with *M. speciosa*. However, their study did not explore the metabolomic variation across species or regions that can indicate any environmental and (or) interspecific influences on the chemical profiles. Similarly, Razafimandimbison and Bremer [6] found that *M. rotundifolia* and *M. diversifolia* exhibited high genetic similarity when both ITS sequencing and morphological traits were analyzed. To our knowledge, this study is the first of its kind to apply chemical separation using the OPLS-DA approach, with results corresponding with morphological and DNA analyses as well as the previously reported literature.

The proposed recursive OPLS-DA approach has significant advantages over conventional one-step binary classification models. By allowing each split to be statistically validated based on *p*-values and metabolite importance, it ensured that only meaningful biological differences were considered, reducing the risk of overfitting or misclassification due to noise on large multi-class datasets. Unlike traditional supervised models that force classifications into predefined categories, this approach enabled a more flexible and data-driven clustering of metabolomic profiles, particularly in cases where class boundaries were not well-defined. The metabolites identified as significant class discriminators at various splits correspond with previously reports, further validating the classification structure and results of the current method as biochemically interpretable. The framework can extend beyond plant metabolomics into broader fields such as microbial metabolomics, clinical biomarker discovery, and environmental chemistry, where recursive splits can indicate the biological or environmental nature of the classification.

## 4. Materials and Methods

### 4.1. Sample Collection and Metabolite Profiling

Sixty-three individual trees (as biological replicates) from the four *Mitragyna* species were sampled across Thailand, spanning diverse climatic zones, and were identified by a taxonomist from Kasetsart University [17]. Leaves from the selected trees were collected from five regions of Thailand (north, east, northeast, central, and south) and were merged into northern and southern regions (Figure 9 and Appendix A). Foliar extracts were prepared following Leksungnoen et al. [34], where dried, ground leaves (20 g) were soaked in methanol (MeOH) for three days, filtered, concentrated using a rotary evaporator, and sonicated before centrifugation. The resulting supernatant was analyzed via GC-MS (QP2020 NX, Shimadzu Corporation, Kyoto, Japan) equipped with an SH-Rxi-5Sil MS column, using helium as the carrier gas at 1 mL/min. Samples were injected in splitless mode, with ion source and interface temperatures set at 250 °C. The mass spectrometer operated at 70 eV electron ionization, scanning *m*/*z* 45–700 at a speed of 2500, with an event time of 0.3 s. The temperature program started at 60 °C, increasing at 8 °C/min to 280 °C, before being held for 25 min.

Chromatograms and mass spectra underwent baseline correction, peak alignment, and normalization before the statistical analysis. Identified compounds were compared against spectral databases, including NIST (https://webbook.nist.gov/chemistry/), MassBank (http://www.massbank.jp accessed on 11 October 2022), and METLIN (http://metlin.scripps.edu/index.php accessed on 11 October 2022), with annotations requiring at least 80% spectral similarity, based on a threshold balancing specificity and sensitivity in metabolomics [35]. While stricter thresholds (>90%) enhance confidence [36], they may exclude low-abundance compounds or those with incomplete reference spectra [37]. Subsequently, the samples were classified into eight classes based on species-level class (four species and two locations).

### 4.2. Statistical Analysis

The foliar metabolome of the four *Mitragyna* species were analyzed using a principal component analysis (PCA), orthogonal partial least squares discriminant analysis (OPLS-DA), hierarchal OPLS-DA, and boxplots. Data normalization via log transformation and Pareto scaling was followed by a fold change analysis (|Log2FC| > 2) and *t*-tests with FDR-adjusted *p*-values less than 0.05 (Benjamini–Hochberg correction [38]). These multivariate analyses were performed to identify key metabolites differentiating the species based on scaled variable importance in projection (VIP) scores. The OPLS-DA was implemented using the *ropls* package in R [39], and model significance was assessed through ten-fold stratified cross-validation to include only those models with a permutation-based *p*-value < 0.05, and metabolites that were significant in discriminating the clusters were selected based on a variable importance in projection (VIP) score > 1 and *p*-value < 0.05 [40]. A hierarchical clustering analysis (HCA), using Ward’s linkage method, was performed to determine the inherent clusters in the metabolome, with the subsequent dendrogram structures visualized using the *factoextra* package in R [41]. The distance matrix was computed using the Euclidean distances between metabolomics samples. Furthermore, boxplots facilitated visualization of interspecific variations in the abundance of metabolites that were deemed as significant discriminators. All the analyses were conducted in the R statistical software version 4.1.1 [42].

### 4.3. Hierarchal Classification on Foliar Metabolome

To resolve the class structure based on biologically interpretable interspecific and regional differences, a recursive OPLS-DA framework was implemented. The methodology integrated unsupervised and automated clustering using t-distributed stochastic neighbor embedding (t-SNE) and k-means, followed by clustering through supervised discrimination using an orthogonal partial least squares discriminant analysis (OPLS-DA) to determine significantly discriminating metabolites. At each recursive split, the non-linear relationships in the foliar metabolomes of the samples were captured using t-SNE (t-distributed stochastic neighbor embedding), which projects the data into a low-dimensional latent space while preserving the global and local structure [43]. Unsupervised clustering performed through t-SNE was implemented with maximum iterations of 10,000, with the resulting three-dimensional t-SNE representation clustered using k-means. Based on the t-SNE representation, low-dimensional embedding was clustered using iterative k-means clustering [44], splitting the data into binary subclasses at each level.

The robustness of each split was quantified using silhouette scores, with a threshold above 0.5 used to retain biologically reliable splits [45]. Only clusters meeting this threshold were analyzed further through OPLS-DA modeling [39]. Importantly, we required that each node of the hierarchy contain a minimum number of samples to enable stratified cross-validation, to avoid overfitting. At each recursive split, the validated OPLS-DA model [46] was used to then identify statistically significant clusters, to progressively refine the class structure through successive OPLS-DA models. The primary advantage of this recursive framework was the elimination of predefined classes (i.e., either species or locations), allowing for a flexible and automated classification structure. The model robustness was first tested on the well-known IRIS dataset, the results of which are discussed in the Appendix A

As mentioned previously, at each significant split, the OPLS-DA model performance was evaluated through 10-fold stratified cross-validation, with a threshold of *p*-values < 0.05 used to indicate statistically significant class separations that were not due to random chance. Metabolites were deemed significant if they had a VIP score greater than 1 and a *p*-value below 0.05 [46]. These metabolites were representative of the influence of either interspecific or environmental factors or both in distinguishing the recursive subclasses found within the metabolome.

This recursive, automated splitting without predefined group labels was used to handle the non-linear and nested variance structures. Compared to a standard pairwise OPLS-DA that could only analyze two classes at a time, this approach can significantly reduce model redundancy, while improving the interpretability of complex metabolomic datasets. The presented recursive approach circumvents this by streamlining classification through data-driven binary classes at each recursive split. All analyses and data visualizations were performed in R [42].

At each recursive split, significant metabolites identified by the OPLS-DA were visualized using S-plots [47], and their differential abundance was subjected to an ANOVA and Tukey’s post hoc tests to determine any significant difference between the two clusters. Additionally, an absolute effect size estimation of the abundance of these metabolites was determined using Cohen’s d values [48] and interpreted using absolute thresholds as either negligible (below 0.2), small (0.2), medium (0.5), or large (0.8).

## 5. Conclusions

The use of a recursive OPLS-DA in handling complex multiclass metabolomic classifications was demonstrated, to overcome the limitations of pairwise binary comparisons in the OPLS-DA. The application of the recursive OPLS-DA on the foliar metabolomes of four *Mitragyna* species demonstrated its ability to identify class structures that were either species-specific, environmental discriminators, or both. Results indicated that the foliar metabolome of *M. speciosa* was distinct from the other three species. On the other hand, the metabolomic profiles of *M. hirsuta* and *M. rotundifolia* were very similar, while *M. diversifolia* exhibited differences that were driven by regional influence caused by environmental differences. These results indicated that the recursive binary classification at each subsequent split was not arbitrary and had biological relevance. Unlike approaches that relied on manual one-vs.-one modeling, the recursive OPLS-DA automated the classification process, reducing bias while preserving the interpretability. As such, the recursive OPLS-DA can be used in metabolomics research, where multi-class datasets frequently exhibit complex, overlapping structures that require adaptive classification strategies. Future work can focus on refining the existing class structure using complementary unsupervised approaches such as a hierarchical clustering analysis (HCA) to validate or augment the splits detected through the recursive OPLS-DA. This would further improve the resolution of biologically meaningful subgroups and interpretability of such complex metabolomic datasets, especially those derived from plants under abiotic stress.

## Figures and Tables

**Figure 1 plants-14-02721-f001:**
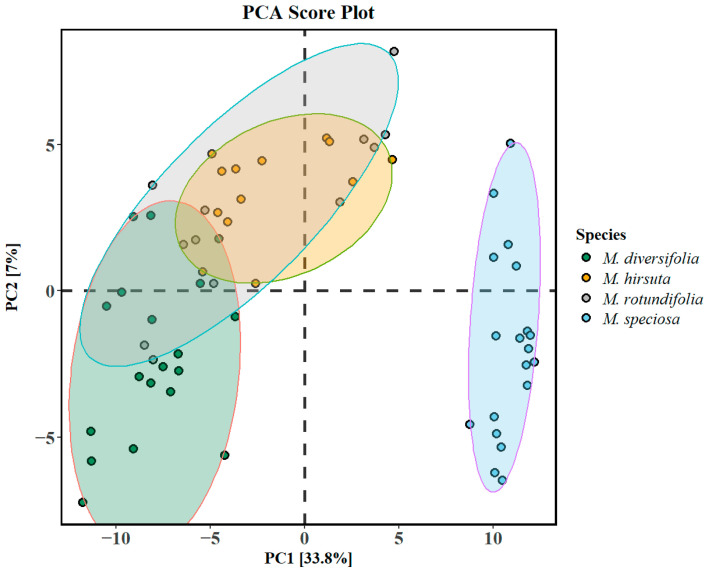
An exploratory principal component analysis (PCA) score plot of the metabolomic distribution in the leaves of the four *Mitragyna* species (shaded ellipses representing 95% confidence intervals) and the variance explained. The four *Mitragyna* species are indicated by abbreviations: MD: *M. diversifolia*; MH: *M. hirsuta* MR: *M. rotundifolia*; MS: *M. speciosa*.

**Figure 2 plants-14-02721-f002:**
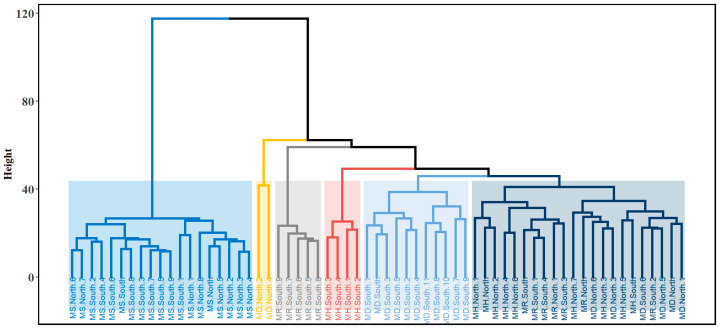
Dendrogram obtained from the hierarchical clustering analysis (HCA) for the foliar metabolomes of the four *Mitragyna* species, as indicated by abbreviations (MD: *M. diversifolia*; MH: *M. hirsuta* MR: *M. rotundifolia*; MS: *M. speciosa*) and locations (north or south). Samples are color-coded by their assigned clusters, based on metabolomic similarities across species and regions. Shaded rectangles indicate cluster boundaries inferred from dendrogram cut points, while the *y*-axis represents the Euclidean distance at which the dendrogram was cut.

**Figure 3 plants-14-02721-f003:**
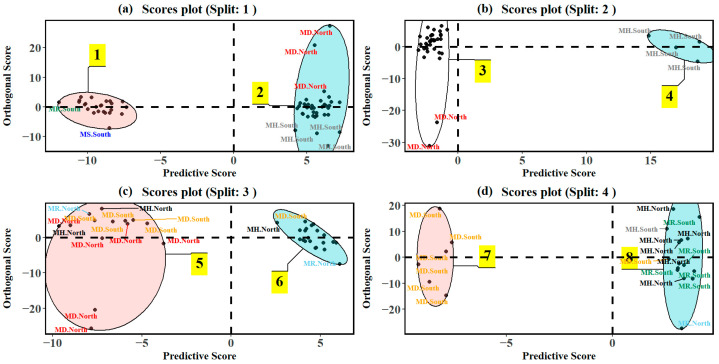
Illustration of the recursive classification of leaf metabolomes of *Mitragyna* species based on eight classes based on sequential binary splits, with samples grouped according to species and geographic origin (panels (**a**)–(**d**)). Ellipses represent 95% confidence intervals, highlighting the metabolic variance within each cluster, while the numbers highlighted in yellow signify clusters to which various samples were designated to according to the hierarchal clustering.

**Figure 4 plants-14-02721-f004:**
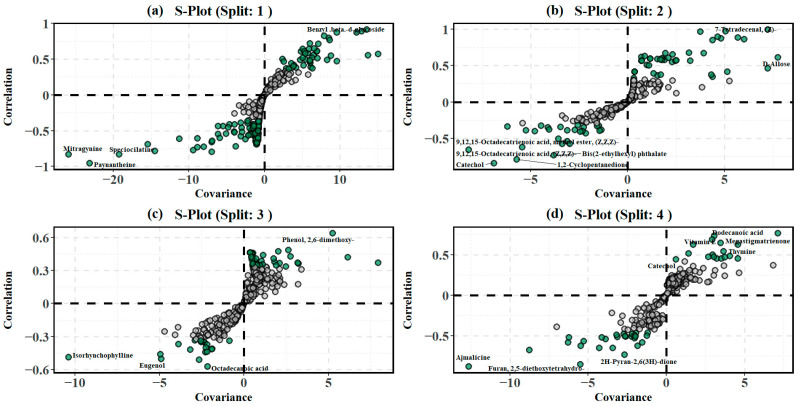
S-plots obtained for the four splits (panels (**a**)–(**d**)) using the recursive OPLS-DA classification of the *Mitragyna* species, highlighting metabolites that were significant in discriminating the two classes (illustrated in green circles, compared to insignificant metabolites plotted in grey circles) built at the four splits through hierarchal clustering. Annotated circles are metabolites of interested in *M. speciosa*.

**Figure 5 plants-14-02721-f005:**
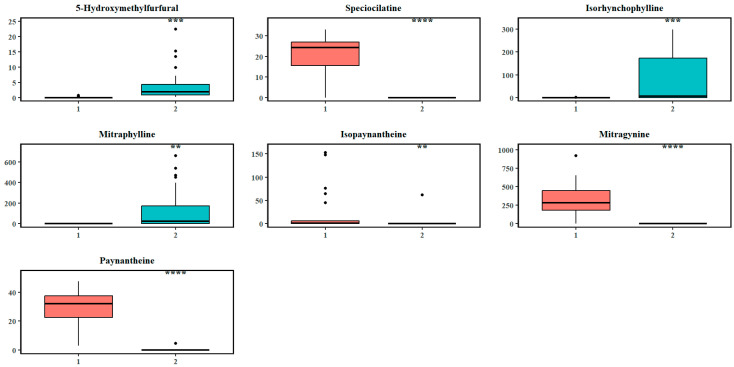
Boxplots of the abundance levels of some metabolites that significantly discriminated the first recursive split (cluster 1: separating all MS trees in the south from the rest of the metabolome; cluster 2: the remaining samples). The circles indicate the outliers in the abundance levels of the respective chemicals. Asterisks denote levels of statistical significance between the two clusters based on hypothesis testing: *p*-value < 0.01 (**), *p*-value < 0.001 (***), and *p*-value < 0.0001 (****).

**Figure 6 plants-14-02721-f006:**
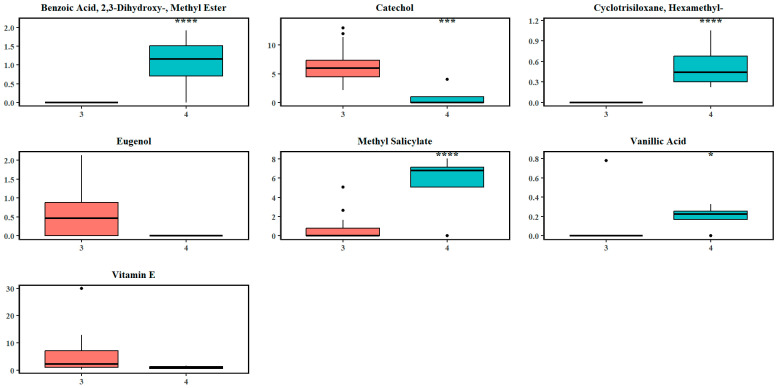
Boxplots of the abundance of some significantly discriminating metabolites at the second recursive split (cluster 3: samples remaining after the first split and not from MH south; cluster 4: MH south samples). The circles indicate the outliers in the abundance of the respective chemicals. Asterisks denote levels of statistical significance between the two clusters based on hypothesis testing: *p*-value < 0.05 (*), *p*-value < 0.001 (***), and *p*-value < 0.0001 (****).

**Figure 7 plants-14-02721-f007:**
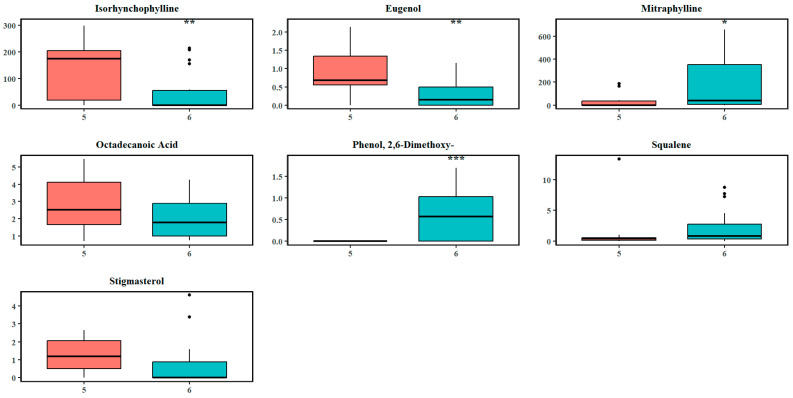
Boxplots of the abundance levels of some significantly discriminating metabolites at the third recursive split. The circles indicate the outliers in the abundance of the respective chemicals. Asterisks denote levels of statistical significance between the two clusters based on hypothesis testing: *p*-value < 0.05 (*), *p*-value < 0.01 (**), and *p*-value < 0.001 (***).

**Figure 8 plants-14-02721-f008:**
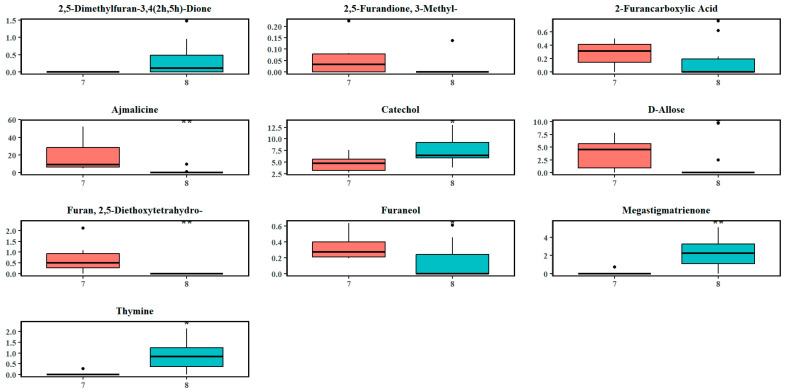
Boxplots of the abundance levels of some significantly discriminating metabolites at the third hierarchical split (cluster 7: most samples from MD south; cluster 8: remaining samples after the first, second, third splits and not from MD south samples). The circles indicate the outliers in the abundance levels of the respective chemicals. Asterisks denote levels of statistical significance between the two clusters based on hypothesis testing: *p*-value < 0.05 (*), and *p*-value < 0.01 (**).

**Figure 9 plants-14-02721-f009:**
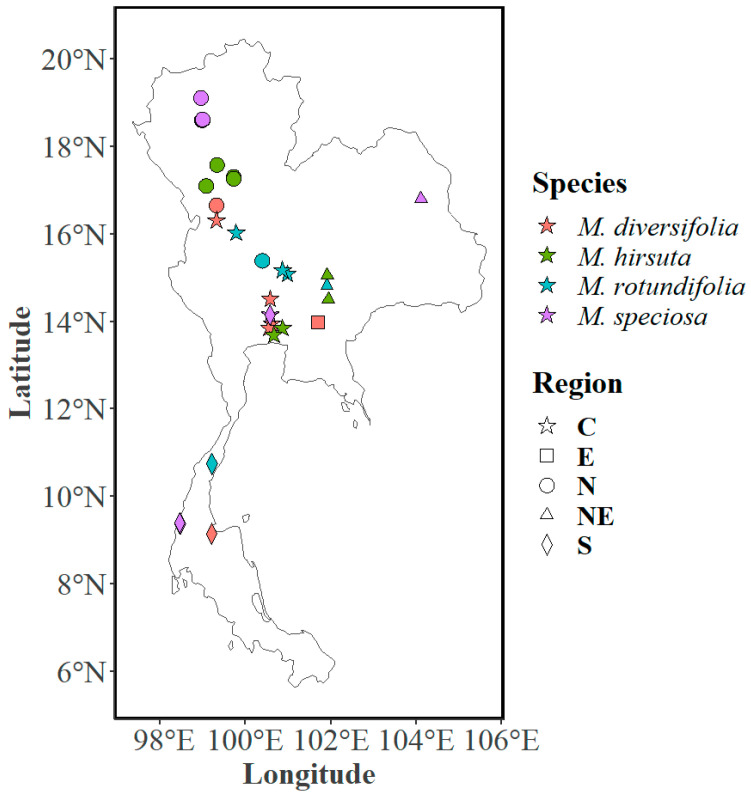
Distribution of the *Mitragyna* species growing in the two regions of Thailand (north and south). The four species are *Mitragyna diversifolia* (Wall. ex G.Don) Havil., *Mitragyna hirsuta* Havil., *Mitragyna rotundifolia* (Roxb.) Kuntze, *and Mitragyna speciosa* (Korth.) Havil. or kratom.

**Table 1 plants-14-02721-t001:** Classification metrics at each recursive split obtained for the *Mitragyna* foliar metabolome.

Split Number	Silhouette Score	OPLS-DA *p*-Value	R2 (Q2)
**1**	0.57	<0.0001	0.98 (0.85)
**2**	0.82	<0.0001	0.96 (0.76)
**3**	0.54	<0.0001	0.91 (0.61)
**4**	0.58	0.002	0.90 (0.58)

## Data Availability

Data used in the analyses can be found in the Appendix A.

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
