# Peer review of "Interspecific and Environmental Influence on the Foliar Metabolomes of Mitragyna Species Through Recursive OPLSDA Modeling"

_plants, 2025, doi:10.3390/plants14172721_

Round 1
Reviewer 1 Report
Comments and Suggestions for Authors
In this manuscript, the authors addressed the limitations of OPLS-DA in multi-class metabolomics data analysis by using a hierarchical OPLS-DA approach. This method integrates a progressive splitting strategy based on hierarchical clustering with OPLS-DA to construct a recursive multi-class discrimination model. While this approach offers promising potential, further validation is required to substantiate its advantages.
Major:
- Provide more detailed methodological explanation of hierarchical OPLS-DA in Method or Result section.
- Perform comprehensive comparison with pairwise OPLS-DA to rigorously demonstrate the advances of hierarchal OPLS-DA. For instance, model performance, analysis efficiency and interpretability.
- The standard approach in metabolomics often combines p-values, fold change, and OPLS-DA VIP scores for identifying differential features. Please compare the hierarchal OPLS-DA with the standard approach.
Author Response
Reviewer 1
Open Review
( ) I would not like to sign my review report
(x) I would like to sign my review report
Quality of English Language
( ) The English could be improved to more clearly express the research.
(x) The English is fine and does not require any improvement.
Yes |
Can be improved |
Must be improved |
Not applicable |
|
Does the introduction provide sufficient background and include all relevant references? |
(x) |
( ) |
( ) |
( ) |
Is the research design appropriate? |
(x) |
( ) |
( ) |
( ) |
Are the methods adequately described? |
( ) |
( ) |
(x) |
( ) |
Are the results clearly presented? |
( ) |
( ) |
(x) |
( ) |
Are the conclusions supported by the results? |
( ) |
( ) |
(x) |
( ) |
Are all figures and tables clear and well-presented? |
( ) |
(x) |
( ) |
( ) |
Comments and Suggestions for Authors
In this manuscript, the authors addressed the limitations of OPLS-DA in multi-class metabolomics data analysis by using a hierarchical OPLS-DA approach. This method integrates a progressive splitting strategy based on hierarchical clustering with OPLS-DA to construct a recursive multi-class discrimination model. While this approach offers promising potential, further validation is required to substantiate its advantages.
Major:
- Provide more detailed methodological explanation of hierarchical OPLS-DA in Method or Result section.
Response: We have added more details about the hierarchical OPLS-DA in the methods section. Also, we have renamed the modeling to recursive OPLS-DA.
- Perform comprehensive comparison with pairwise OPLS-DA to rigorously demonstrate the advances of hierarchal OPLS-DA. For instance, model performance, analysis efficiency and interpretability.
Response: We started the metabolomic analysis using the usual suite of tools that also included OPLS-DA. The pairwise OPLS-DA analyses was conducted across all the regional and interspecific combinations. Out of 28 pairwise comparisons, only two models showed statistically significant discrimination (p < 0.05), while the majority were either statistically insignificant. This motivated us to build the hierarchical OPLS-DA framework, which improved model significance by not just trying to classify prescribed classes but the class structure was based on meaningful classes in the multiclass metabolomic dataset. In other words, the hierarchical OPLS-DA framework enabled discovery of treatment-driven class structures and provided interpretable, non-redundant groupings. The pairwise model diagnostics are summarized in Supplementary Table S3.
- The standard approach in metabolomics often combines p-values, fold change, and OPLS-DA VIP scores for identifying differential features. Please compare the hierarchal OPLS-DA with the standard approach.
Response: We appreciate the reviewer’s point regarding the standard statistics used in metabolomics. As indicated in the previous response, most pairwise OPLS-DA models were statistically insignificant, limiting the reliability of a conventional p-value and VIP-based feature selection for discriminating metabolites. For consistency with standard metabolomics standards, we have now included p-values and VIP scores for discriminant metabolites at each hierarchical split in Supplementary Tables labeled Splot_metabs_Split1 to 4.csv. Furthermore, as seen in Table S2, out of 28 pairwise comparisons, only two models were statistically significant (p < 0.05), while the majority were statistically insignificant.
- Figures and tables can be improved
Response: We have added more figures as per suggestions from the other reviewers as well as tables and tried to improve the existing ones. These include an unsupervised PCA has been added, cluster numbers have been added in recursive OPLS-DA alongwith further statistics regarding the four splits, boxplots indicating abundance of significant metabolites in clusters at each split and previously reported in Mitragyna species.
Reviewer 2 Report
Comments and Suggestions for Authors
The original article deals with metabolic profile of Mitargina plant species that grow under different environment conditions in Thailand. Various species of the genus Mitragyna have become popular phytotherapy, especially for the inhibition of pain. Therefore, the topic of the manuscript is interesting and important in the scientific field.
The analytical method, gas chromatography-mass spectrometry, was adequate. The consequent statistical evaluation offered interesting results, which supports conclusions. Additionally, the results are presented as illustrative figures and properly discussed. Furthermore, the authors cited the most relevant references.
Author Response
Reviewer 2
Open Review
(x) I would not like to sign my review report
( ) I would like to sign my review report
Quality of English Language
( ) The English could be improved to more clearly express the research.
(x) The English is fine and does not require any improvement.
Yes |
Can be improved |
Must be improved |
Not applicable |
|
Does the introduction provide sufficient background and include all relevant references? |
(x) |
( ) |
( ) |
( ) |
Is the research design appropriate? |
(x) |
( ) |
( ) |
( ) |
Are the methods adequately described? |
(x) |
( ) |
( ) |
( ) |
Are the results clearly presented? |
(x) |
( ) |
( ) |
( ) |
Are the conclusions supported by the results? |
(x) |
( ) |
( ) |
( ) |
Are all figures and tables clear and well-presented? |
(x) |
( ) |
( ) |
( ) |
Comments and Suggestions for Authors
The original article deals with metabolic profile of Mitargina plant species that grow under different environment conditions in Thailand. Various species of the genus Mitragyna have become popular phytotherapy, especially for the inhibition of pain. Therefore, the topic of the manuscript is interesting and important in the scientific field.
The analytical method, gas chromatography-mass spectrometry, was adequate. The consequent statistical evaluation offered interesting results, which supports conclusions. Additionally, the results are presented as illustrative figures and properly discussed. Furthermore, the authors cited the most relevant references.
Response: We thank the reviewer for the positive evaluation and appreciate the recognition of the relevance of our study on Mitragyna species under varying environmental conditions, as well as the appropriateness of our analytical methods, statistical interpretation, and presentation of results. We are also grateful for the acknowledgment of the clarity of our discussion and the use of relevant literature.
Reviewer 3 Report
Comments and Suggestions for Authors
The manuscript employs a method integrating hierarchical clustering and OPLS-DA analysis to investigate the disparities in leaf metabolite data across four Mitragyna species, with the objective of shedding light on the influence of distinct genetic or environmental factors on metabolites. Nevertheless, the manuscript predominantly centers on the efficacy of the method utilized, neglecting to delve into the biological differences among the various Mitragyna species. As a result, it fails to present other crucial data findings. It is advisable that the authors undertake substantial revisions to the manuscript prior to considering its acceptance. The specific issues are detailed below:
-2.1: The utilization of the IRIS dataset by the authors to verify the accuracy of their novel method is not suitable, given that this pre-existing dataset primarily comprises phenotypic data and lacks the complexity associated with metabolite data. In my opinion, this step is redundant since the authors proceed to analyze the data from the current study regardless.
-There is an absence of descriptions concerning fundamental quality control and descriptive statistical outcomes for the data generated via GC-MS. This renders the analytical results of the manuscript somewhat superficial and makes it challenging for readers to ascertain the reliability of the data. It is highly recommended that the authors incorporate pertinent basic data analyses, such as the classification of identified metabolites, their variations within the same species, and results from PCA, HCA, OPLS-DA, among others. Moreover, they should compare these results with those derived from their hierarchical clustering-OPLS-DA analysis to enhance the presentation of data results and validate that their method outperforms conventional techniques.
-Do the scores plots in Figures 4, 5, 6, and 7 replicate the scores plot in Figure 3? This raises significant questions regarding the rationale behind such duplication. It is suggested that the Results section be restructured and additional figures and tables be produced to elucidate the core themes more effectively.
-Discussion and Conclusion: Following the revision of the Results section, corresponding amendments should also be implemented in the Discussion and Conclusion sections.
-Methods Section: The number of biological replicates tested must be explicitly stated. Additionally, fundamental details such as the common names of all compounds and their database matching scores should be included. Furthermore, it is recommended to append the total ion chromatograms (TIC) from the GC-MS detection of these eight species to facilitate a more intuitive comparison of their metabolite differences.
-The figure numbers require correction.
-Throughout the manuscript, including in the references, the Latin scientific names of species should be italicized.
Author Response
Reviewer 3
Open Review
( ) I would not like to sign my review report
(x) I would like to sign my review report
Quality of English Language
( ) The English could be improved to more clearly express the research.
(x) The English is fine and does not require any improvement.
Yes |
Can be improved |
Must be improved |
Not applicable |
|
Does the introduction provide sufficient background and include all relevant references? |
( ) |
(x) |
( ) |
( ) |
Is the research design appropriate? |
( ) |
(x) |
( ) |
( ) |
Are the methods adequately described? |
( ) |
( ) |
(x) |
( ) |
Are the results clearly presented? |
( ) |
( ) |
(x) |
( ) |
Are the conclusions supported by the results? |
( ) |
( ) |
(x) |
( ) |
Are all figures and tables clear and well-presented? |
(x) |
( ) |
( ) |
( ) |
Comments and Suggestions for Authors
The manuscript employs a method integrating hierarchical clustering and OPLS-DA analysis to investigate the disparities in leaf metabolite data across four Mitragyna species, with the objective of shedding light on the influence of distinct genetic or environmental factors on metabolites. Nevertheless, the manuscript predominantly centers on the efficacy of the method utilized, neglecting to delve into the biological differences among the various Mitragyna species. As a result, it fails to present other crucial data findings. It is advisable that the authors undertake substantial revisions to the manuscript prior to considering its acceptance. The specific issues are detailed below:
-2.1: The utilization of the IRIS dataset by the authors to verify the accuracy of their novel method is not suitable, given that this pre-existing dataset primarily comprises phenotypic data and lacks the complexity associated with metabolite data. In my opinion, this step is redundant since the authors proceed to analyze the data from the current study regardless.
Response: We acknowledge the concern regarding the use of the IRIS dataset for verification. The intention behind using this standard dataset was to perform a model verification and sanity check to ensure the robustness of the OPLS-DA method employed. Specifically, the IRIS dataset, with its well-documented structure, contains the Setosa species, which is clearly separated from the other two classes (Versicolor and Virginica). This characteristic makes it suitable for benchmarking classification performance.
Furthermore, the same dataset has been used in a recent study by Forsgren et al. (2024), in which the a similar separation structure was observed as in the current study. In other words, our results using the same dataset demonstrated a comparable separation pattern, which supports the validity of the OPLS-DA implementation for further use in characterizing the foliar metabolome of the Mitragyna species. However, in response to the reviewer’s suggestion and to avoid redundancy in the main text, we have now moved the IRIS dataset validation and related figures to the Supplementary Material section. We hope this adjustment addresses the reviewer’s concern and clarifies the rationale for using the IRIS dataset to verify the modeling scheme presented in the current study.
Forsgren, Edvin, Benny Bjorkblom, Johan Trygg, and Par Jonsson. "OPLS-based multiclass classification and data-driven interclass relationship discovery." Journal of Chemical Information and Modeling 65, no. 4 (2025): 1762-1770.
-There is an absence of descriptions concerning fundamental quality control and descriptive statistical outcomes for the data generated via GC-MS. This renders the analytical results of the manuscript somewhat superficial and makes it challenging for readers to ascertain the reliability of the data. It is highly recommended that the authors incorporate pertinent basic data analyses, such as the classification of identified metabolites, their variations within the same species, and results from PCA, HCA, OPLS-DA, among others. Moreover, they should compare these results with those derived from their hierarchical clustering-OPLS-DA analysis to enhance the presentation of data results and validate that their method outperforms conventional techniques.
Response: We appreciate the reviewer’s valuable feedback highlighting the need for more comprehensive quality control and descriptive statistical analyses to support the reliability of our GC-MS metabolomic data. To address these points thoroughly, we have added the following in the revised manuscript:
- An unsupervised PCA has been added to visualize overall data structure.
- Cluster numbers have been added in the recursive OPLS-DA results, with extended statistics for the four hierarchical splits.
- Boxplots illustrating the abundance of significant metabolites across clusters (at each split) have been introduced, including metabolites previously reported in Mitragyna species.
Comparison to Conventional Methods
- As noted in our earlier response, pairwise OPLS-DA models across regional and interspecific combinations largely failed to achieve statistical significance (as summarized in Supplementary Table S2), motivating the development of our recursive OPLS-DA framework.
- We’ve ensured consistency with standard metabolomics benchmarks by providing p-values and VIP scores for discriminant metabolites at each split (available in Supplementary Tables Splot_metabs_Split1.csv through Split4.csv).
-Do the scores plots in Figures 4, 5, 6, and 7 replicate the scores plot in Figure 3? This raises significant questions regarding the rationale behind such duplication. It is suggested that the Results section be restructured and additional figures and tables be produced to elucidate the core themes more effectively.
Response: We acknowledge the reviewer’s observation regarding the apparent redundancy of the scores plots across Figures 3 through 7 (the figure numbers are now different in the current version). The rationale behind this structure was to first establish the complete recursive structure of the results obtained from the hierarchical OPLS-DA model, which indicated four statistically significant splits in the metabolome. Figures 4 to 7 then revisit these same score plots individually to allow focused interpretation at each split, accompanied by the corresponding S-plots of metabolites which were annotated using VIP scores and p-value to now with the inclusion of boxplots exhibiting relative abundance of some metabolites previously reported in these species. This layered presentation was intended to guide the reader progressively from a broader model structure at each split to metabolite-level interpretation without packing the visuals in a single figure. While the score plots were reused, their contextual framing was different, making them integral to understanding both the modeling perspective as well as the biological interpretations at each split.
-Discussion and Conclusion: Following the revision of the Results section, corresponding amendments should also be implemented in the Discussion and Conclusion sections.
Response: Relevant information has been added based on additional figures as per the suggestion.
-Methods Section: The number of biological replicates tested must be explicitly stated. Additionally, fundamental details such as the common names of all compounds and their database matching scores should be included. Furthermore, it is recommended to append the total ion chromatograms (TIC) from the GC-MS detection of these eight species to facilitate a more intuitive comparison of their metabolite differences.
Response: We appreciate the reviewer’s suggestion and acknowledge the importance of transparent compound identification in metabolomics studies. The study included 63 biological replicates, as detailed in the referenced article (Leksungnoen et al. 2022). We have included the common names that were found for annotated metabolites, as indicated in the supplementary Excel file (Table S1). We would also like to correct the confusion of the reviewer when they say there were eight species. The number of species was four, which were sampled from locations in the northern and the southern parts of Thailand, hence resulting in eight classes.
Regarding similarity scores, we have included a representative metadata file containing information about the foliar metabolome of a single tree named MS south (MS1(1).xlsx) as obtained from the original vendor-exported file and included as a supplementary file. These include retention time (RT), area, height, and observed m/z, alongwith the TIC plot etc. The MS data used to annotate each compound using built-in GC-MS software tools and cross-referenced mass spectral libraries (e.g., NIST), typically used in compound annotation are also indicated.
An example of this structure is provided in the uploaded file where compounds are matched based on retention index alignment and spectral similarity. Given the scale of the dataset (63 GC-MS chromatograms across four Mitragyna species), individual total ion chromatograms (TICs) for each run were not included as this would not offer additional discriminative insights beyond the statistical analyses already performed.
Leksungnoen, Nisa, Tushar Andriyas, Chatchai Ngernsaengsaruay, Suwimon Uthairatsamee, Phruet Racharak, Weerasin Sonjaroon, Roger Kjelgren, Brian J. Pearson, Christopher R. McCurdy, and Abhisheak Sharma. "Variations in mitragynine content in the naturally growing Kratom (Mitragyna speciosa) population of Thailand." Frontiers in plant science 13 (2022): 1028547.
-The figure numbers require correction.
Response: The figure numbers have been corrected according to the order in which they are referenced in the text.
-Throughout the manuscript, including in the references, the Latin scientific names of species should be italicized.
Response: We have checked for and corrected the names as suggested.
Round 2
Reviewer 1 Report
Comments and Suggestions for Authors
The authors have addressed all my questions and I am satisfied with the current version.
Author Response
Reviewer 1
The authors have addressed all my questions and I am satisfied with the current version.
Response: We thank the reviewer for their time and feedback, and we appreciate the acknowledgment of various revisions made.
Reviewer 3 Report
Comments and Suggestions for Authors The authors have made effective revisions to the manuscript, particularly by adding some fundamental descriptions and analyses regarding VOCs. In the subsequent revised manuscript, it is recommended that the authors highlight the modified sections for the reviewers to assess the revisions. There are still some issues in the manuscript that require the authors' attention: -The authors have added some descriptions. However, fundamental descriptions of VOCs remain inadequate. For instance, what categories (alcohols, esters, etc.) do the 409 detected VOCs belong to? How many VOCs are there in each category? Which VOCs have higher relative contents? Additionally, it is recommended to supplement the Hierarchical Cluster Analysis (HCA) to observe the relationships among samples and among VOCs, providing a more intuitive representation of the data in Table S1. -If Figures 3, 5, 7, and 9 are redundant with the scores plot in Figure 2, it is, in my opinion, unacceptable. Since your data have already been presented once in Figure 2, repeating them later with minor adjustments can easily mislead readers into thinking they are from other analyses. Perhaps you could integrate the four S-plot figures into a new Figure 3, which would be more in line with the requirements of scientific papers and more concise. Moreover, it avoids increasing the manuscript's length with redundant figures. -The authors should align the files "MS1(1).xlsx" and "Table S1.csv" in the supplementary materials. The compounds in "Table S1.csv" only have relative contents but lack VOC names, while the "MS1(1).xlsx" file only has compound names without corresponding contents. Furthermore, why do the numbers of VOCs in the two files not match? Additionally, supplementary materials should be uniformly named as Table SX. -The thresholds for Medium and Small in the Magnitude of effect size in Table S3 need to be added to the title notes. -The file names should correspond to the titles within the files. The title in the Table S3 file is Table S2, whereas the title in the Table S2 file is Table S3.Author Response
Reviewer 3
Comments and Suggestions for Authors
The authors have made effective revisions to the manuscript, particularly by adding some fundamental descriptions and analyses regarding VOCs. In the subsequent revised manuscript, it is recommended that the authors highlight the modified sections for the reviewers to assess the revisions.
Response: We appreciate the reviewer’s acknowledgment of the revisions made. We would like to however clarify that the overarching focus of the current study was on recursive classification of foliar secondary metabolites using untargeted GC-MS-based metabolomic profiling through location- and species-level classification.
If the reviewer is referring to volatile organic compounds (VOCs), we clarify that VOC profiling was not within the current scope of this work. Metabolite relative abundance data were processed without assigning compounds to specific volatile categories. While common names of metabolites are included in the supplementary file S1, chemical class annotations and other metrics (e.g., category-level accumulation or TIC overlays) were not included apart from certain metabolites such as Mitragynine, whose accumulation and variation has already been reported in a previous paper (Leksungnoen et al. 2022). Nonetheless, we acknowledge that integrating VOC analysis could be a valuable future investigation to complement the recursive classification approach introduced here.
In the revised manuscript, all modified sections have been clearly highlighted in yellow to facilitate an efficient review of the edits made.
Leksungnoen, Nisa, Tushar Andriyas, Chatchai Ngernsaengsaruay, Suwimon Uthairatsamee, Phruet Racharak, Weerasin Sonjaroon, Roger Kjelgren, Brian J. Pearson, Christopher R. McCurdy, and Abhisheak Sharma. "Variations in mitragynine content in the naturally growing Kratom (Mitragyna speciosa) population of Thailand." Frontiers in plant science 13 (2022): 1028547
There are still some issues in the manuscript that require the authors' attention:
The authors have added some descriptions. However, fundamental descriptions of VOCs remain inadequate. For instance, what categories (alcohols, esters, etc.) do the 409 detected VOCs belong to? How many VOCs are there in each category? Which VOCs have higher relative contents?
Additionally, it is recommended to supplement the Hierarchical Cluster Analysis (HCA) to observe the relationships among samples and among VOCs, providing a more intuitive representation of the data in Table S1.
Response: We would like to clarify that the current study focused on untargeted GC-MS-based metabolomics of foliar secondary metabolites, and not on volatile organic compounds (VOCs). If the reviewer is referring to VOCs, we note that no VOCs were classified in this study, as this was not the interest of the current study. Consequently, quantification or categorization of VOCs was not conducted. Additionally, since this was an untargeted metabolomics study, relative accumulation levels of the 409 compounds were not compared in quantitative terms, except for mitragynine, which was addressed in a prior study focused on its accumulation across populations in Thailand (Leksungnoen et al., 2022). We again agree that targeted VOC profiling would add further value to the current study through a future research.
As recommended, a Hierarchical Cluster Analysis (HCA) dendrogram has now been included (Figure 2) to illustrate an unsupervised perspective on sample similarity. While the current figure focuses on sample-wise clustering of the foliar metabolome, feature-wise (VOC-level as suggested) clustering could shed light on the underlying metabolite level co-regulation and will be considered in a future analyses.
Leksungnoen, Nisa, Tushar Andriyas, Chatchai Ngernsaengsaruay, Suwimon Uthairatsamee, Phruet Racharak, Weerasin Sonjaroon, Roger Kjelgren, Brian J. Pearson, Christopher R. McCurdy, and Abhisheak Sharma. "Variations in mitragynine content in the naturally growing Kratom (Mitragyna speciosa) population of Thailand." Frontiers in plant science 13 (2022): 1028547.
If Figures 3, 5, 7, and 9 are redundant with the scores plot in Figure 2, it is, in my opinion, unacceptable. Since your data have already been presented once in Figure 2, repeating them later with minor adjustments can easily mislead readers into thinking they are from other analyses. Perhaps you could integrate the four S-plot figures into a new Figure 3, which would be more in line with the requirements of scientific papers and more concise. Moreover, it avoids increasing the manuscript's length with redundant figures.
Response: We have now consolidated the four S-plots into a single figure (now Figure 3), by the respective split number, as per the suggestion.
The authors should align the files "MS1(1).xlsx" and "Table S1.csv" in the supplementary materials. The compounds in "Table S1.csv" only have relative contents but lack VOC names, while the "MS1(1).xlsx" file only has compound names without corresponding contents. Furthermore, why do the numbers of VOCs in the two files not match?
Response: We appreciate the reviewer’s attention to detail. The focus of the present study was not on VOC ontologies or categorization, but rather on developing a recursive multiclass classification framework to explore any metabolite-level differences across samples. Therefore, the supplementary files were intended to serve different roles—“MS1(1).xlsx” documents compound-level identifications, while “Table S1.csv” provides the associated presence of the metabiolites used in recursive OPLS-DA clustering. The aim was to identify statistically significant metabolites at each recursive split, contributing to class separation based on treatment and species-level factors. As such, ontology-level summaries (e.g., VOC categorization by chemical class or compound-level accumulation) were not included. Nonetheless, we again agree that this addition could be a valuable extension in future work for additional biological interpretation.
Additionally, supplementary materials should be uniformly named as Table SX.
Response: The material has been renamed as suggested. Also, we have added a list of all the material supplied with the manuscript in a document titled “List of material.docx”.
The thresholds for Medium and Small in the Magnitude of effect size in Table S3 need to be added to the title notes. Response: We have updated the caption of Table S3 to now include the following note for clarity: "The interpretation of estimates are based on absolute thresholds of negligible (d < 0.2), small (d > 0.2), medium (d > 0.5), or large (d > 0.8)." as per the suggestion.
The file names should correspond to the titles within the files. The title in the Table S3 file is Table S2, whereas the title in the Table S2 file is Table S3.
Response: All file names and corresponding captions have now been corrected as per the suggestion.
Round 3
Reviewer 3 Report
Comments and Suggestions for Authors The author has basically resolved the major issues in the manuscript and addressed my doubts. -Line 439: It should be specified that the "factoextra package" is an R package, and relevant references need to be added. Additionally, clarification is required regarding the distance measure used for HCA. Is it Euclidean distance? Moreover, it would be preferable to add an axis indicating the distance on the HCA plot in Figure 2.Author Response
General Response to Editor and Reviewers
We are again sincerely grateful to the editor and reviewers for their valuable feedback, which has significantly improved the overall quality of the manuscript.
Reviewer 3
The author has basically resolved the major issues in the manuscript and addressed my doubts.
Response: We are glad that the revisions have addressed the concerns and appreciate the inputs throughout the review process.
-Line 439: It should be specified that the "factoextra package" is an R package, and relevant references need to be added.
Response: We have now specified the appropriate reference for the factoextra package and indicated that it is an R package.
Additionally, clarification is required regarding the distance measure used for HCA. Is it Euclidean distance?
Response: The reviewer is correct in pointing out that the Hierarchical Cluster Analysis (HCA) was performed through the distance matrix computed using Euclidean distance. This detail has now been added in the revised Methods section.
Moreover, it would be preferable to add an axis indicating the distance on the HCA plot in Figure 2.
Response: The revised HCA plot (Figure 2) now includes a y-axis, which has been added to the figure caption.